# Primary Cutaneous Rhabdomyosarcomatous Melanomas—A Report of Two Cases and Literature Review

**DOI:** 10.3390/diagnostics15111357

**Published:** 2025-05-28

**Authors:** Andreea Iliesiu, Victor Nimigean, Dana Antonia Tapoi, Mariana Costache

**Affiliations:** 1Department of Pathology, Carol Davila University of Medicine and Pharmacy, 050474 Bucharest, Romania; andreea.iliesiu@umfcd.ro (A.I.); mariana.costache@umfcd.ro (M.C.); 2Department of Pathology, University Emergency Hospital, 050098 Bucharest, Romania; 3Department of Anatomy, Faculty of Dentistry, Carol Davila University of Medicine and Pharmacy, 050474 Bucharest, Romania; 4Department of Pathology, Synevo Romania, 077040 Bucharest, Romania

**Keywords:** melanoma, dedifferentiated, rhabdoid, rhabdomyosarcomatous, immunohistochemistry, genetic mutations

## Abstract

**Background and clinical significance:** Cutaneous melanomas sometimes display unusual histopathological features, reminiscent of various other malignancies, either primary or metastatic. However, due to the highly aggressive nature of cutaneous melanomas, an accurate and timely diagnosis is mandatory. This requires extensive histopathological and immunohistochemical analyses and molecular tests, if needed. **Case presentation:** In this respect, we present two cases of primary cutaneous melanomas exhibiting rhabdoid features and genuine divergent rhabdomyosarcomatous differentiation confirmed by immunoreactivity for myogenin and loss of positivity for some melanocytic markers. We discuss the diagnosis approach for these particularly rare entities, highlighting the most useful immunohistochemical panel. Additionally, we also provide an extensive review of all the previously reported similar lesions, focusing on the epidemiological, histopathological, immunohistochemical and molecular features, as well as discussing the prognostic and treatment options for rhabdomyosarcomatous cutaneous melanomas. **Conclusions:** These rare cases of primary cutaneous melanomas with rhabdomyosarcomatous differentiation underscore the diagnostic challenges posed by such unusual histopathological variants. In order to establish the correct diagnosis a comprehensive immunohistochemical workup, including both melanocytic and myogenic markers, is required. These findings are supported by a detailed review of the literature, emphasizing the importance of recognising these rare melanoma subtypes for providing the appropriate prognostic assessment, and therapeutic management.

## 1. Introduction

Melanoma is a heterogeneous disease characterised by significant variability across clinical, histopathological, immunohistochemical, and molecular features, all of which influence both therapeutic strategies and prognosis [1]. A subset of cutaneous melanomas can undergo divergent differentiation, displaying immunohistochemical and ultrastructural features of other cell lineages such as epithelial cells, fibroblasts, nervous cells, osteocartilaginous cells, endothelial cells, smooth muscle, or rhabdomyoblasts [2,3,4]. In 1992, Bittesini L et al. first described a rhabdoid melanoma in lymph node metastasis [5]. To date, several other cases of rhabdoid melanomas have been reported but most of them involve metastatic lesions [6,7,8,9,10,11,12,13,14,15,16,17,18,19,20,21,22]. However, not all the cases displayed definitive rhabdomyosarcomatous dedifferentiation, confirmed by immunohistochemical analysis or electron microscopy studies. Additionally, rhabdomyosarcomatous dedifferentiation has also been described in recurrent melanomas [23,24].

On the contrary, primary rhabdoid melanomas are exceedingly rare. Nevertheless, these features have been reported in both primary mucosal melanoma [25,26,27,28,29,30] and primary cutaneous melanoma [30,31,32,33,34,35,36,37,38,39,40,41,42]. Notably, the majority of these cases fail to exhibit true immunohistochemical myogenic differentiation, with only a limited number of primary cutaneous melanomas displaying desmin positivity [30,33,37,40]. Furthermore, primary cutaneous melanomas with authentic rhabdomyoblastic differentiation demonstrated by myoglobin immunoreactivity remain exceptionally uncommon [30,40]. In this article, we present two additional cases of primary cutaneous melanoma with rhabdomyoblastic differentiation and provide a comprehensive review of the previously reported rhabdoid melanomas, both primary or metastatic. Emphasis is placed on cases with genuine rhabdomyosarcomatous dedifferentiation by discussing the epidemiology, diagnostic approaches, and prognosis of these patients.

## 2. Materials and Methods

This study was approved by the Ethics Committee at the University Emergency Hospital of Bucharest, Romania. This study followed the principles of the Helsinki Declaration. Additionally, both patients included in this article signed an informed consent form.

The tissue samples obtained from the surgical removal of the cutaneous tumours were fixed in 10% buffered histologic formalin, paraffin-embedded, sectioned, and Hematoxylin–Eosin (HE)-stained according to conventional histology methods.

Immunostaining was performed using the Dako labeled Streptavidin–Biotin (LSAB) (Agilent Technologies, Inc., Santa Clara, CA, USA) kit with 3,3′-diaminobenzidine as chromogen. The following antibodies were provided by Biocare (Biocare Medical, Pacheco, CA, USA): HMB45 (mouse monoclonal, clone HMB-45, 1:100 dilution), SOX10 (mouse monoclonal, clone BC34, Biocare Clone, 1:100 dilution), Prame (rabbit monoclonal, clone EPR20330), 1:100 dilution, Vimentin (mouse monoclonal, clone V9, 1:200 dilution), Desmin (mouse monoclonal, clone D33, 1:50 dilution) and Myogenin (mouse monoclonal, clone MyG007—also known as F5D, 1:100 dilution). Genetic testing for *BRAF* mutations was performed using an Idylla™ *BRAF* Mutation Assay cartridge (Biocartis, Mechelen, Belgium).

Additionally, an extensive review of the literature was conducted, encompassing papers published up to the year 2025 in PubMed-indexed journals that discuss melanomas with rhabdoid features, with emphasis on those demonstrating true rhabdomyosarcomatous differentiation. All types of articles were included: reviews, original studies, and case reports. The keywords used in the database search were rhabdoid melanoma, rhabdomyosarcomatous melanoma, and rhabdomyoblastic melanoma.

Descriptive statistics were provided for continuous variables, including mean and standard deviation (SD). Comparisons for continuous variables were performed using the unpaired *t*-test while the Chi-Square test was used for categorical variables. Statistical significance was established at a *p*-value threshold of <0.05. Statistical analysis was performed using GraphPad Prism 10.4.2 (Graphpad Software Inc., San Diego, CA, USA).

## 3. Results

### 3.1. Case 1

A 41-year-old male presented to the Emergency University Hospital of Bucharest in 2023, following an episode of an epileptic seizure. The patient had no notable medical history. The only significant finding revealed by the clinical examination was an ulcerated nodular lesion measuring 8 × 7 × 5 cm with adjacent confluent nodules with dimensions of 9 × 4 × 2.5 cm on his left thigh (Figure 1).

The tumour was surgically removed with wide resection margins (>2 cm) and sent for histopathological and immunohistochemical examination.

Histopathological sections showed a proliferation of highly atypical neoplastic cells arranged in nests and lobules separated by fibrous septa. The tumour exhibited ulceration of the overlying epidermis and extended deeply into the subcutaneous tissue (Figure 2).

The tumour cells were large and polygonal, with an abundant eosinophilic cytoplasm containing hyaline inclusions and large, eccentric nuclei with prominent nucleoli. Multinucleate cells were also present. Additional noteworthy findings included areas of necrosis and vascular emboli. Based on these features, the initial differential diagnoses were alveolar soft part sarcoma and rhabdomyosarcoma. Nevertheless, as more sections were evaluated, a few scattered pigmented cells were noticed, thus raising the suspicion of melanoma with rhabdoid features (Figure 3).

Immunohistochemical analysis revealed the tumours cells to be completely negative for HMB45. In contrast, there was diffuse positivity for desmin and focal positivity for myogenin (Figure 4).

However, the tumour cells displayed diffuse positivity for SOX10 and Prame, including in the rhabdoid areas (Figure 5). Based on these results, the diagnosis of primary cutaneous melanoma with rhabdomyosarcomatous differentiation was established.

Following the diagnosis, genetic tests for BRAF mutations revealed a wild-type phenotype. The patient also underwent full-body CT scans and brain MRI, which demonstrated the presence of cerebral metastases. The patient was initially lost to follow-up but returned to the hospital twelve months later with ascites, breathing difficulties, and altered mental state. A novel full-body CT scan revealed an increase in the cerebral metastasis as well as the presence of abdominal and pulmonary metastases. The patient died due to widespread disease 12 months later.

### 3.2. Case 2

A 42-year-old male presented to the Emergency University Hospital of Bucharest in 2024, with an ulcerated nodular lesion measuring 6 × 5.5 × 2.8 cm with extensive areas of necrosis and haemorrhage on his posterior thorax. The patient also reported worsening fatigue and persistent back pain over the preceding several months. The tumour was surgically removed with wide resection margins (>2 cm) and sent for histopathological and immunohistochemical examination. The clinical exam revealed no other skin lesions.

Histologically, the tumour was composed of nests and sheets of large pleomorphic cells, ulcerating the epidermis and invading the skeletal muscle of the thorax. The tumour cells were large and polygonal with an abundant eosinophilic cytoplasm. The nuclei were eccentric and had a conspicuous nucleolus (Figure 6).

Additionally, extensive areas of necrosis were noted, and following extensive analysis of multiple sections, a few scattered pigmented cells were revealed (Figure 7).

Immunohistochemically, the cells were completely negative for HMB45, while being focally positive for both desmin and myogenin (Figure 8).

Nevertheless, the tumour cells were diffusely positive for SOX10 and Prame (Figure 9). Based on these features, the diagnosis of primary cutaneous melanoma with rhabdomyosarcomatous differentiation was established.

Following this diagnosis, genetic tests for BRAF mutations were again performed and no abnormalities were noted in this regard. The patient also underwent full-body CT scans and brain MRI which revealed the presence of cerebral, pulmonary, and bone metastases. The patient initially received anti-PD1 therapy, but a new full body CT scan three months later showed an increase in the number and dimension of the metastatic lesions. Consequently, chemotherapy with dacarbazine was started. Nevertheless, subsequent CT scans at 6 months after the diagnosis revealed the disease had continued to progress and the patient succumbed to the disease 10 months later.

## 4. Discussion

Rhabdoid morphology in melanomas was originally described in a metastatic lesion in 1992 [5] and to this date, although uncommon, several other authors reported similar features in metastatic melanoma [6,7,8,9,10,11,12,13,14,15,16,17,18,19,20,21,22]. Nevertheless, rhabdoid morphology does not inherently imply true rhabdomyoblasic differentiation. In light of this view, we compare the main histological and immunohistochemical features observed in previously reported metastatic rhabdoid melanomas (Table 1).

The mean age of the patients presented in Table 1 was 54.2 (SD = 18.08) and the male to female ratio was 1:1.5. Interestingly, only ten cases exhibited genuine rhabdomyosarcomatous dedifferentiation proved by myogenin or myoglobin immunopositivity. In this group, the mean age of the patients was 50.64 (SD = 19.11) and the male to female ratio was 1:4.5.

Regarding primary cutaneous melanomas with rhabdoid features, the previously reported cases are presented in Table 2.

The mean age of the patients presented in Table 2 is 75.75 (SD = 9.49) and the male to female ratio was 2.375:1. However, only 11 of these cases displayed genuine rhabdomyosarcomatous dedifferentiation. The demographic and clinical characteristics of these patients together with the two new cases with primary cutaneous melanomas with rhabdomyosarcomatous dedifferentiation are presented in Table 3.

Considering only the previously reported cases, the mean age of the patients was 73.91 (SD = 14.06) and the male-to-female ratio was 2.67:1. Interestingly, the two new cases presented in this paper further confirm the tendency of rhabdomyosarcomatous melanomas to affect males more frequently than females, but the age of these two patients was considerably younger than the mean age of all the previously reported cases. By also considering these two cases, the mean age of patients with primary rhabdomyosarcomatous melanomas was 68.92 (SD = 17.41), and the male-to-female ratio was 3.33:1. Furthermore, as shown previously, dedifferentiated melanomas tend to occur on highly sun-exposed skin [3], which may explain the older age of presentation. In this respect, seven of the previously reported rhabdomyosarcomatous melanomas occurred on the head and neck. However, the two new cases presented in this study did not arise from highly sun-damaged skin.

Regarding the prognosis of primary dedifferentiated melanomas, it is generally considered poor [3], and these two new cases we reported confirm this hypothesis. Nevertheless, both cases presented with locally advanced disease and their reduced survival may be merely the consequence of the delayed diagnosis, as Breslow depth remains the most important prognostic factor in cutaneous melanomas [50]. In comparison, four of the previously reported rhabdomyosarcomatous melanomas died of the disease with a mean survival time of just 5.75 months. Overall, 61.54% of all the primary rhabdomyosarcomatous melanoma patients progressed to metastatic disease, the lungs being involved in each case. The mean age of the patients who developed metastases was 59 (SD = 15.06), while the mean age of the patients without metastases was 88 (SD = 5.72). This difference was highly significant (*p* = 0.0045; unpaired *t*-test). Nevertheless, these findings may not accurately reflect the behaviour of rhabdomyosarcomatous melanomas due to the low number of cases and because, in four of the remaining seven cases, there was either no follow-up or a very short follow-up period.

In comparison, metastatic rhabdomyosarcomatous melanomas affected significantly younger individuals (*p* = 0.0251; unpaired *t*-test) and female patients more frequently than male patients (*p* = 0.0041; Chi-Square test). These differences further highlight the highly heterogeneous behaviour of cutaneous melanomas. While the exact mechanisms of dedifferentiation in primary cutaneous melanoma remain unknown, this phenomenon, when encountered in metastatic lesions, may represent a way of gaining resistance to therapy [51,52].

Finally, in extremely rare instances, rhabdomyosarcomatous features have been described in melanomas arising in congenital nevi (Table 4) [53,54].

However, only one of the two cases of congenital melanomas exhibited true rhabdomyosarcomatous dedifferentiation, as confirmed by immunohistochemical positivity for myogenin and myoD1. Having considered this case, as well as the two newly reported cases and the eleven previously reported cases of primary rhabdomyosarcomatous melanomas, these entities remain extraordinarily rare, thus posing significant diagnostic challenges. A melanoma diagnosis can be easily considered when an in situ or conventional melanoma area is present, but, just as in our cases, these features may not always be evident. Hence, extensive immunohistochemical studies are required, using a broad panel of melanocytic markers. In this regard, conventional melanocytic markers such as HMB45, MelanA, or SOX10 are most often negative in dedifferentiated melanomas [3]. The expression of these markers is also lost in the majority of the rhabomyosarcomatous melanomas. S100 demonstrates greater sensitivity, with 50% of the previously reported rhabdomyosarcomatous melanomas retaining its expression. Nevertheless, its diagnostic utility is limited by its lack of specificity, as S100 is also positive in other neoplasms such as MPNST, which can further complicate the definitive diagnosis [55]. Prame appears to be the most sensitive marker for dedifferentiated melanomas, as described in previous studies [3]. Regarding rhabdomyosarcomatous melanomas, Prame expression was retained in all four cases evaluated to date, which comprise the two new cases reported in this article and two previously published cases. However, due to the limited number of analysed samples and because Prame can also be expressed in other tumours, further studies are required to fully assess its sensitivity and specificity in diagnosing rhabdomyosarcomatous melanomas.

Finally, molecular tests may be useful for establishing the correct diagnosis by identifying genetic mutations that are characteristic of cutaneous melanomas. As dedifferentiated melanomas frequently occur on highly sun-damaged skin, NF1 mutations are the most commonly encountered [3]. Considering the rhabdomyosarcomatous melanomas, molecular data were available for eight of the previously reported cases. Of these, the most frequently mutated characteristic gene was indeed NF1, identified in three cases, followed by NRAS in two cases. Interestingly, TP53 was also mutated in three cases and CDK2NA and TERT were mutated in two cases. Even though these alterations are not characteristic for cutaneous melanomas, recognizing them is particularly important for prognostic evaluations. TP53, CDKN2A, and TERT promoter mutations are usually associated with advanced disease [56] which is consistent with the presumed poor prognosis of rhabdomyosarcomatous melanomas. Nevertheless, identifying these promoter mutations may also influence the therapeutic approach. Patients harbouring TERT promoter or CDKN2A alterations may benefit from targeted therapy [57]. Lastly, BRAF mutations, considered the most common in cutaneous melanoma and significant therapeutic targets [57], remain rare in dedifferentiated lesions, as none of the rhabdomyosarcomatous melanomas harboured this modification. These findings are also confirmed by the two cases presented in this article, as neither of them was BRAF mutated. On the contrary, the mutational landscape of metastatic rhabdomyosarcomatous melanomas was quite different. Of the nine metastatic cases with available molecular profiling, BRAF mutations were identified in seven, while NF1 was mutated in only one patient and no NRAS mutations were noted. This observation further underscores the unpredictable nature of rhabdomyosarcomatous melanomas, whether primary or metastatic. In this context, further studies are necessary to develop the best diagnostic and therapeutic strategies for these rare entities.

In regard to the treatment of dedifferentiated cutaneous melanomas, the management of the patients varies based on the stage of the disease. The tumours must be surgically removed with peripheral resection margins ranging from 0.5 mm for in situ lesions to >2 mm for tumours thicker than 2 mm. Sentinel lymph node biopsy should be performed for melanomas thicker than 1 mm or 0.8 mm with additional risk factors [58]. The follow-up guidelines differ based on country, but they generally include CT/PET-CT and brain MRI every 3–6 months for patients with stage IIB/IIC or higher [59]. At present, adjuvant chemotherapy is generally recommended for patients with stage III melanoma (patients with positive lymph nodes) as well as for patients with metastatic disease [60]. Additionally, the FDA has also approved treatment for stage IIB//IIC [61]. The most widely used adjuvant and sometimes neoadjuvant drugs include target therapy with BRAF/MEK inhibitors as well as immunotherapy with PD-1 inhibitors or CTLA-4 inhibitors [60]. Regarding dedifferentiated cutaneous melanomas, there are currently no specific treatment guidelines available. As these neoplasms tend to be BRAF non-mutated, they are most often treated with immune checkpoint inhibitors. Additionally, other therapeutic options may include conventional chemotherapy and radiation therapy [24]. Radiotherapy may be used for treating the primary tumour in patients too frail to undergo surgical excision or for brain and bone metastases. Conventional chemotherapy is rarely used for treating cutaneous melanomas nowadays but may still be used as a last resort option for BRAF wild-type cases who develop resistance or severe toxicity to immunotherapy [58]. Despite this remarkable progress in treating cutaneous melanomas, the rate of resistance to the novel therapies remains high. As a consequence, new drugs are being developed. At present, clinical trials have shown good response rates for anti-VEGF molecules, oncolytic viral therapy, and T-cell antagonists [60]. Due to the great variety of novel therapies, further studies are required to fully evaluate the best options for dedifferentiated cutaneous melanomas.

## 5. Conclusions

Melanomas with genuine rhabdomyosarcomatous dedifferentiation, either primary or metastatic, are exceedingly rare tumours. By presenting two new cases of primary cutaneous rhabdomyosarcomatous melanomas, we further documented the epidemiology, diagnostic approach, and prognostic significance of these entities. Due to the rarity of such lesions, extensive immunohistochemical and molecular tests may be required for accurate diagnosis.

## Figures and Tables

**Figure 1 diagnostics-15-01357-f001:**
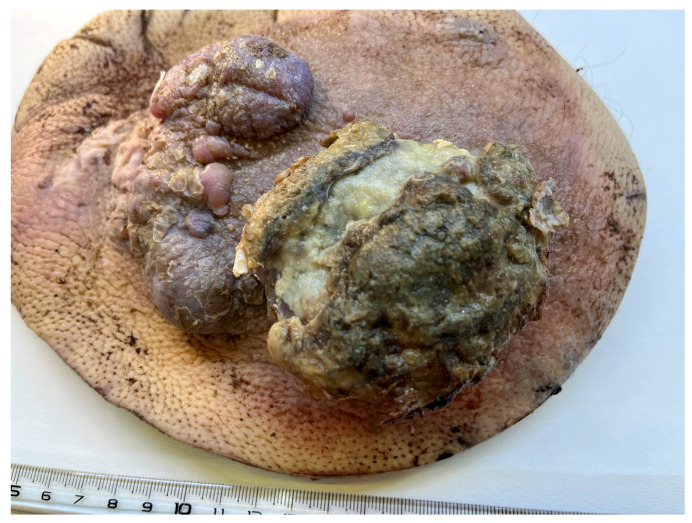
Large, ulcerated nodular tumour with smaller satellite nodules.

**Figure 2 diagnostics-15-01357-f002:**
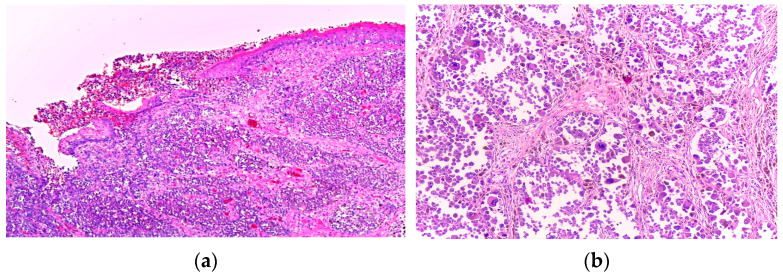
(**a**) Nests of tumour cells ulcerating the epidermis (H&E, 4×); (**b**) Highly pleomorphic cell nests separated by fibrous septa (H&E, 20×).

**Figure 3 diagnostics-15-01357-f003:**
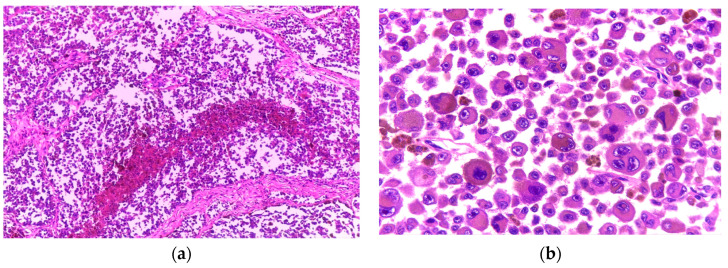
(**a**) Tumour cells in a pseudo-alveolar growth pattern with central discohesion and necrosis (H&E, 10×); (**b**) Large, pleomorphic cells with eosinophilic cytoplasm and eccentric nuclei. Occasional multinucleated and pigmented cells are noted (H&E, 40×).

**Figure 4 diagnostics-15-01357-f004:**
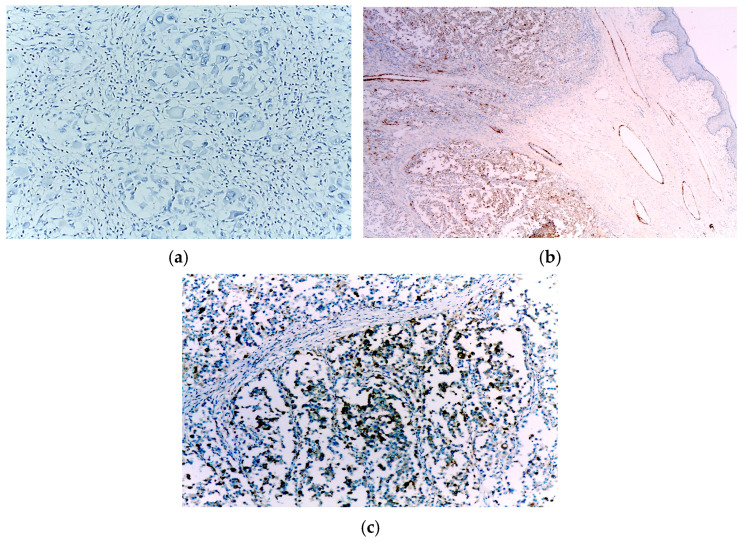
(**a**) Negative HMB45 immunoexpression in the tumour cells; (**b**) Diffuse positivity for desmin in the tumour cells; (**c**) Focal positivity for myogenin in the tumour cells.

**Figure 5 diagnostics-15-01357-f005:**
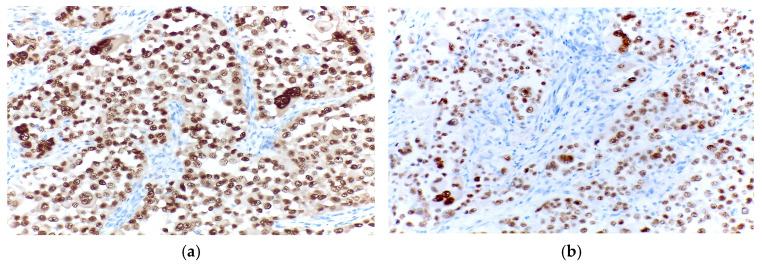
(**a**) Diffuse positivity for SOX10 in the tumour cells; (**b**) Diffuse positivity for Prame in the tumour cells.

**Figure 6 diagnostics-15-01357-f006:**
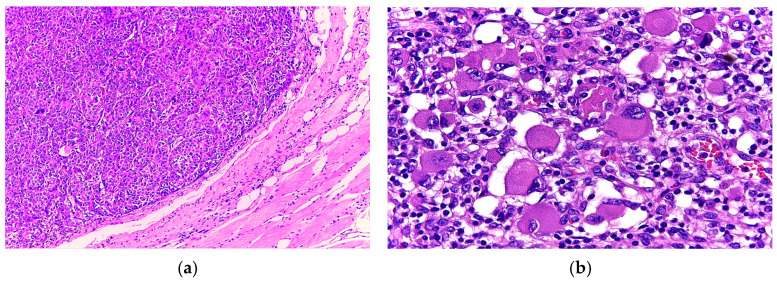
(**a**) Nests of tumour cells infiltrating the skeletal muscle (H&E, 4×); (**b**) Rhabdoid cells with abundant eosinophilic cytoplasm and eccentric nuclei (H&E, 40×).

**Figure 7 diagnostics-15-01357-f007:**
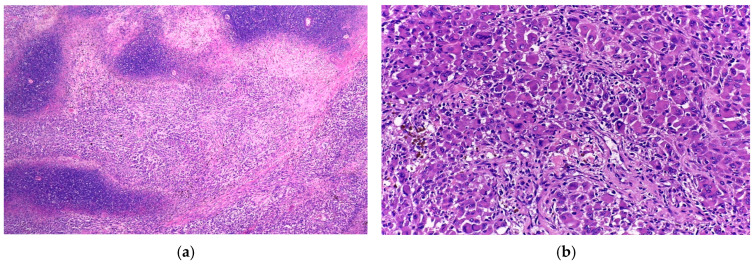
(**a**) Extensive areas of geographic necrosis intermingled with sheets of tumour cells (H&E, 4×); (**b**) Sheets of large rhabdoid cells and a few, smaller pigmented cells (H&E, 20×).

**Figure 8 diagnostics-15-01357-f008:**
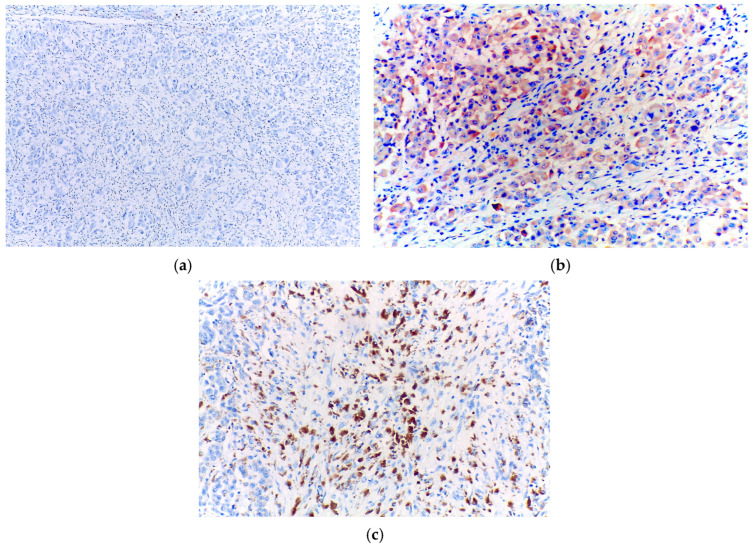
(**a**) Negative HMB45 immunoexpression in the tumour cells; (**b**) Focal positivity for desmin in the tumour cells; (**c**) Focal positivity for myogenin in the tumour cells.

**Figure 9 diagnostics-15-01357-f009:**
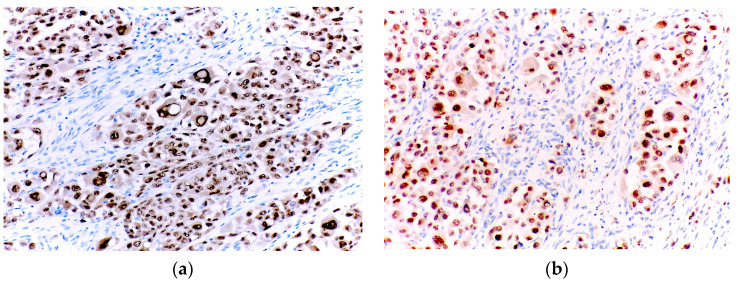
(**a**) Diffuse positivity for SOX10 in the tumour cells; (**b**) Diffuse positivity for Prame in the tumour cells.

**Table 1 diagnostics-15-01357-t001:** Previously reported cases of metastatic melanomas with rhabdoid features.

Reference	Age	Gender	Primary Tumour	Site	Melanocytic Markers	Other Markers	Ultrastructural Features	Molecular Tests
Bittesini L et al., 1992 [5]	74	Male	N/A	Axillary lymph nodes	Negative: S100, HMB45	Positive: Vimentin, desmin	Whorls of intermediate filaments; no melanosomes	NP
Laskin WB et al., 1995 [6]	60	Male	Melanoma NOS	Subcutaneous tissue	Negative: S100, HMB45	Positive: vimentin, desmin Negative: myoglobin	Paranuclear aggregates of intermediate filaments; no tonofilaments, melanosomes, or skeletal muscle differentiation.	NP
Pasz-Walczak G et al., 2002 [7]	32	Female	Melanoma NOS	Lymph node	Positive: S100 Negative: HMB45	Positive: desmin	NP	NP
	41	Female	Melanoma NOS	Brain	Positive: HMB45	Positive: desmin	NP	NP
Abbott JJ et al., 2004 [8]	62	Male	N/A	Lung	Positive: S100, focally Mart-1 Negative: HMB45, MelanA, tyrosinase	Positive: Vimentin, CD56 Negative: desmin, SMA, EMA	Paranuclear whorls of intermediate filaments and no melanomas	NP
Gharpuray-Pandit D et al., 2007 [9]	21	Female	N/A	Latero-cervical lymph nodes	Focally positive: S100, HMB45, MelanA, Tyrosinase	Positive: Desmin, myoglobin, myogenin and Myo D1 in rhabdoid areas	Nonpigmented melanosomes with primitively developed lattices; no rhabdomyoblastic differentiation	NP
Gavino AC et al., 2008 [10]	64	Male	Acral melanoma	Subcutaneous tissue	Focally positive: S100, HMB45	Positive: Vimentin Negative: desmin, SMA, MSA	Condensed, twisted sheaves of intermediate filaments	NP
Nakamura T et al., 2009 [11]	44	Female	N/A	Subcutaneous tissue	Positive: 100, MelanA Negative: HMB45	Positive: Vimentin, EMA Negative: Desmin, CD34, cytokeratin	NP	NP
Guo R et al., 2013 [12]	85	Male	Nodular Melanoma	Parotid gland	Positive: MIFT Negative: S100, HMB45, MelanA, Tyrosinase	Desmin-, SMA-	NP	gains in 6p (RREB1), chromosome 7, and 8q11.1-q24.3 (MYC) additional copy of chromosomal 7q (BRAF), loss of chromosome 9p24.3-q13 (CDKN2A) and chromosome 4 (Melan A encoding gene)
Reilly DJ et al., 2014 [13]	59	Male	Nodular Melanoma	Axillary lymph nodes	S100, HMB45, MelanA, Tyrosinase focally positive, negative in rhabdoid areas	Positive: Desmin, myogenin Negative: SMA	NP	BRAF mutated
Agaimy A et al., 2016 [14]	24	Female	Melanoma NOS	Lung	Positive in isolated cells: S100, HMB45, PanMelanoma Negative: HMB45, MelanA	Positive: desmin, SMA, myogenin, myoglonin	NP	BRAF V6000E mutated
	69	Female	N/A	Jejunum	Negative: S100, HMB45, MelanA, PanMelanoma, SOX10	Positive: desmin, myoglobin, myogenin Negative: SMA	NP	BRAF wild-type, NRAS wild-type
Dumitru AV et al., 2018 [15]	57	Female	N/A	Ileum	Positive: PanMelanoma, MART-1, S100 in classic areas Negative: HMB45	Positive in rhabdoid areas: Desmin, actin, myoglobin	NP	NP
Campbell K et al., 2018 [16]	52	Female	Invasive melanoma	Axillary lymph nodes	Positive: S100, Mart-1	Positive: desmin Negative: myogenin	NP	BRAF V6000E mutated
				Thoracic vertebra	Focally positive: HMB45 Negative: S100, Mart-1, SOX10	Positive: desmin, myogenin	NP	NP
				Kidney	Focally positive: HMB45, SOX10, Mart-1 Negative: S100	Positive: desmin, myogenin	NP	NP
Tran TAN et al., 2019 [17]	65	Female	Nodular Melanoma	Lung	Negative: S100, SOX10, MITF, MelanA, HMB45	Positive: desmin, myogenin	NP	BRAF V600E mutated
Mehta A et al., 2020 [18]	65	Male	N/A	Lung	Positive: HMB45, MelanA, SOX10 Negative: S100	Positive: Vimentin, desmin, SMA, WT1	NP	BRAF wild-type, c-KIT mutated
Tzanavaris K et al., 2022 [19]	63	Male	Nodular Melanoma	Base of tongue	Positive: S00, HMB45, MelanA, SOX10	Focally positive: Myogenin, Myo-D1, chromogranin and synaptophysin	NP	BRAF V600E mutated
Cilento MA et al., 2022 [20]	57	Female	Melanoma NOS	Abdominal	Focal positivity for melanocytic markers	Positive: Myogenin	NP	BRAF V600E mutated
Kasago IS, 2023 [21]	69	Female	N/A	Deep abdominal wall	Focally positive: Prame Negative: S100, HMB45, SOX10, MelanA,	Positive: Desmin, Myogenin, MyoD1	NP	NF1 R1241 mutated
	21	Female	Melanoma NOS		Positive: Prame, BRAF V600E Negative: S100, HMB45, SOX10, MelanA	Positive: Desmin, Myogenin, MyoD1	NP	BRAF V600E mutated

N/A = not available; NP = not performed.

**Table 2 diagnostics-15-01357-t002:** Previously reported cases of primary cutaneous melanomas with rhabdoid features.

Reference	Age	Gender	Conventional Melanoma	Site	Immunohistochemistry	Molecular Tests	Treatment and Follow-Up
Parham DM et al., 1994 [31]	7	Male	N/A	Back	Positive: S100, vimentin	NP	9 months
Borek BT et al., 1998 [32]	30	Female	Nodular melanoma	Thigh	Positive: S100, vimentin Negative: HMB45, SMA, desmin	NP	60 months
	36	Female	Nodular melanoma	Scalp	Positive: S100, vimentin Negative: HMB45, SMA, desmin	NP	12 months
	59	Male	Nodular melanoma	Back	Positive: S100, vimentin, SMA Negative: HMB45, desmin	NP	N/A
Gattenlöhner S et al., 2008 [33]	41	Male	Primary cutaneous melanoma	Scalp	Positive: S100, MelanA, desmin, myogenin	Loss of chromosome 1q31, amplification of 1q32, gain of 12q23-qter	6 months: DOD
Tallon B et al., 2009 [34]	74	Male	Primary cutaneous spindle cell melanoma	Back	Positive: S100, HMB45, MelanA in spindle cells, negative in rhabdoid cells, Negative: desmin	NP	12 months: alive with pulmonary metastases
Chung BY et al., 2011 [35]	54	Male	Primary cutaneous melanoma	Forearm	Positive: S100, HMB45, Fontana-Masson silver, vimentin Negative: CD68, CD34, CD99, SMA, desmin	NP	4 months: no progression
Shenjere P et al., 2014 [30]	67	Female	Primary cutaneous melanoma	Chest	Positive: S100, HMB45, MelanA, MITF in conventional melanoma; Desmin, myogenin, myoD1 in rhabdoid areas	BRAF wild type	2 years: pulmonary metastases but died from unrelated causes
Kaneko T et al., 2015 [36]	63	Male	Primary cutaneous melanoma	Heel	Positive: S100, MelanA, HMB45, vimentin Negative: desmin, αSMA	BRAF wild type	Chemotherapy + immunotherapy 44 months: pulmonary metastases
Fernández-Vega I et al., 2016 [37]	80	Male	Primary cutaneous melanoma	Forehead	Weakly positive: S100, SOX10 Positive: vimentin, desmin Negative: HMB45, MelanA, SMA, myoD1	NP	Radiotherapy 2 months: DOD
Antonov NK et al., 2016 [38]	75	Male	Primary cutaneous melanoma	Scalp	Focally positive: MelanA, S100, myogenin Positive: desmin	NP	Chemotherapy 7 months: DOD
Prieto-Torres L et al., 2016 [39]	69	Female	Primary cutaneous melanoma	Scapula	Positive: S100, SOX10, desmin, SMA Negative: MITF, MelanA, HMB45, MyoD1, myogenin, myoglobin	NP	14 months: no progression
Kuwadekar A et al., 2018 [40]	72	Male	Superficial spreading melanoma	Scalp	Positive: desmin, myogenin Negative: S100, SOX10, HMB45, MelanA	NP	N/A
Murakami T et al., 2019 [41]	78	Male	Primary cutaneous melanoma	Forehead	Positive: S100, vimentin Negative in rhabdoid cells: MelanA, HMB45 Negative: desmin	NP	24 months: no progression
Tran TAN et al., 2019 [42]	96	Male	Lentigo maligna melanoma	Forearm	Positive: desmin, myoD1, myogenin Negative: S100, SOX10, HMB45, MelanA	NRAS c.182A, KDR c.3434G	5 months: local recurrences
Ferreira I et al., 2021 [43]	68	Male	Superficial spreading melanoma	Nose	Positive: desmin, myogenin, myoD1 Negative: S100, SOX10, HMB45, MelanA	NF1, TP53, CDKN2A, RAC1	8 months: DOD
	85	Female	Desmoplastic melanoma	Chin	Positive: des-min, myogenin, myoD1 Negative: S100, SOX10, HMB45, MelanA	NF1, TP53, ATRX, RASA2	34 months: no progression
Agaimy A et al., 2021 [44]	55	Female	N/A	Lower leg	Negative: S100, SOX10, HMB45, MelanA, Pan-Melanoma	BRAF V600E	Inguinal, subcutaneous lung metastasis; no follow-up
Yim SH et al., 2022 [45]	64	Male	Melanoma arising in nevus	Scalp	Positive: desmin, myoD1 Weakly positive: S100, SOX10 Negative: HMB45, MelanA, BRAF V600E	NP	Chemotherapy 2 months: DOD
Glutsch V et al., 2022 [46]	72	Female	Acral lentiginous melanoma	Ankle	Positive: S100, SOX10, MART1, HMB45, vimentin, PRAME Negative: desmin	NP	Chemotherapy 11 months: DOD
	74	Male	Nodular melanoma	Chest	Positive: desmin, PRAME Unspecific/focally positive: SOX10, HMB45 Negative: S100, MART1	NP	2 months: DOD
	75	Male	Nodular melanoma	Scalp	Positive: S100, SOX10, MART1, HMB45, vimentin, PRAME Negative: desmin	NP	3 months: in transit metastasis
	79	Male	Nodular melanoma	Arm	Positive: S100, SOX10, vimentin, PRAME Negative: desmin, MART1, HMB45	NP	N/A
O’Neill P et al., 2023 [47]	74	Male	Nodular melanoma	Chest	Positive: desmin, myoD1, myogenin Negative: SOX10, HMB45, MelanA	NRAS, TERTp, CDKN2A, NF1, FGFR2, CBL, BLM and TP53	Immunotherapy 42 months: widespread metastasis
Choy A et al., 2025 [48]	88	Male	Desmoplastic melanoma	Scalp	Focally positive: SOX10, S100, PRAME, MelanA, desmin, MyoD1 Negative: AE1/AE3, SMA, ERG, BRAF V600E	N/A	2 months: no progression
Weigelt MA et al., 2025 [49]	83	Female	N/A	Deltoid region	Positive: PRAME, desmin, myogenin, myoD1 Negative: cytokeratins, HMB45, S100, SOX10, BRAF V600E, NRAS Q61R	ARID2, BRCA2, CRKL, FANCB, LZTR1, TERT, APC, EGFR, ERBB2, FLCN, TP53	Immunotherapy 10 months: no progression

N/A = not available; NP = not performed.

**Table 3 diagnostics-15-01357-t003:** Primary cutaneous melanomas with rhabdomyosarcomatous dedifferentiation.

Reference	Age	Gender	Tumour Site	Metastatic Site
Gattenlöhner S et al., 2008 [33]	41	Male	Scalp	Lung, mediastinum, abdominal organs
Shenjere P et al., 2014 [30]	67	Female	Chest	Lung
Antonov NK et al., 2016 [38]	75	Male	Scalp	Lung
Kuwadekar A et al., 2018 [40]	72	Male	Scalp	N/A
Tran TAN et al., 2019 [42]	96	Male	Forearm	Local recurrence
Ferreira I et al., 2021 [43]	68	Male	Nose	Lung, brain
	85	Female	Chin	No progression
Yim SH et al., 2022 [45]	64	Male	Scalp	Lung
O’Neill P et al., 2023 [47]	74	Male	Chest	Lung, liver, spleen, bone
Choy A et al., 2025 [48]	88	Male	Scalp	No progression
Weigelt MA et al., 2025 [49]	83	Female	Deltoid region	No progression
This paper	41	Male	Thigh	Lung, brain, abdominal organs
	42	Male	Back	Lung, brain, bone
Summary	Mean: 68.92 (SD:17.41)	76.92% male (*n* = 10) 23.08% female (*n* = 3)	53.84% head and neck (*n* = 7) 23.08% trunk (*n* = 3) 23.08% limbs (*n* = 2)	Metastatic cases: 66.67% (*n* = 8) Matastatic sites: 100% lung (*n* = 8), 37,5% abdominal organs (*n* = 3), 37,5% brain (*n* = 3), 20% bone (*n* = 2)

**Table 4 diagnostics-15-01357-t004:** Previously reported cases of rhabdoid melanomas arising in congenital nevi.

Reference	Age	Gender	Site	Immunohistochemistry	Molecular Tests	Follow-Up
Koyama T et al., 1996 [53]	Newborn	Female	Scalp	S100, HMB45, NKI/C3 positive in conventional area, negative in dedifferentiated area Desmin, sarcomeric actin positive in dedifferentiated area Negative: myoglobin, myosin, SMA	NP	N/A
Baltres, A et al., 2019 [54]	15 months	Female	Lumbosacral	Negative: SOX10, MiTF, HMB45, MelanA Positive: myoD1, myogenin, desmin	SASS6-RAF1 fusion	9 months: lung and liver metastasis

N/A = not available; NP = not performed.

## Data Availability

This article does not include any additional primary data besides the information already presented in the case report section.

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
