# Peer review of "Primary Cutaneous Rhabdomyosarcomatous Melanomas—A Report of Two Cases and Literature Review"

_diagnostics, 2025, doi:10.3390/diagnostics15111357_

Round 1
Reviewer 1 Report
Comments and Suggestions for Authors
This manuscript reports two new cases of primary cutaneous melanoma with true rhabdomyosarcomatous differentiation. Through histopathological, immunohistochemical, and molecular analyses, the authors compare it with previously published cases. An extensive literature review is provided, contextualizing the findings in terms of diagnosis, immunophenotype, and prognosis which is very nice to read
overall the topic provide a rare tumor type which is not well known and the review section cover most important published articles
however, i have some comments:
1-The manuscript requires careful English editing. Numerous grammatical errors and awkward phrasings
2- similarity index : 23% , need to be <15%
3- the format is not acceptable in a case series format, and should be a case report
the minimum recommended number of case series is usually 4-5 (it should be a case report and literature review). Even better to consider a systematic review of case reports
4- i'm not sure if we can perform statistical tests such as chi square/ t-test in review literature
that's why i was wondering if it's better to be a case report + systematic review
5- in case if statistical test such is considered, maybe it will be nice to search the association with metastasis and <65 years >65 years (not only gender)
6- a table gathering demographic data will be also nice including current 2 presented cases (mean (SD) for age. , number of males /females with %). sites (number and percentage), number of metastasis and location (or at least a pie chart of locations) but i extremely recommend a table.
7-clinical image of the tumor could be also nice to add if you have it
8- adding treatment done for each reviewed article is important in my views (surgical treatment/ chemotherapy/ radiotherapy etc)
9- case presentation (of 2 cases), add size of lesion , workup for staging (what imaging), how follow-up was done, surgical margins (R classifications)
10- enrich discussion with staging work-up imaging, and recommended type of follow up (local only/ PET/ or CT chest/ local MRI ?) , minimal recommended surgical margins, recommended treatment (surgical only / chemotherapy in metastatic cases or other )
overall i believe that this study merit publication, but it needs some cleaning and revision with previous comments
Comments on the Quality of English Languageminor english editing
similarity index should be decreased
Author Response
We would like to thank the reviewer for reviewing this manuscript and for subsequent suggestions. We have carefully adressed each one of them.
Comment 1: The manuscript requires careful English editing. Numerous grammatical errors and awkward phrasings
Response 1: Thank you for this suggestion. We have asked an English professor to help review the English.
Comment 2: similarity index : 23% , need to be <15%
Response 2: This is an important observation. We could not detect this index with the programme we used but we tried to reduce it even more by rephrasing, adding more paragraphs and tables according to your suggestions. We hope the result has improved.
Comment 3: the format is not acceptable in a case series format, and should be a case report
Response: Thank you for this suggestion. We have modified the title to better express the purpose of this manuscript.
Comment 4: i'm not sure if we can perform statistical tests such as chi-square/ t-test in review literature
that's why i was wondering if it's better to be a case report + systematic review
Response 4: We appreciate this valuable feedback. As mentioned before we changed the title of the manuscript. We believe that due to the metholodology and relatively low number of cases, a systematic review would be difficult to conduct. We have published literature reviews before in high-impact journal with similar statistical analysis and as we mentioned throughtout the manuscript, the readers should keep in mind that the results are based on a small number of cases due to the rarity of this tumour.
Comment 5: in case if statistical test such is considered, maybe it will be nice to search the association with metastasis and <65 years >65 years (not only gender)
Response 5: This is a very good suggestion. We have added an analysis for the age (lines 351-353).
Comment 6: a table gathering demographic data will be also nice including current 2 presented cases (mean (SD) for age. , number of males /females with %). sites (number and percentage), number of metastasis and location (or at least a pie chart of locations) but i extremely recommend a table.
Response 6: We appreciate this recommendation and believe that the table you mentioned would significantly increase the quality of the manuscript. As a consequence, we provived a new table (Table 3) to include the requested data.
Comment 7: clinical image of the tumor could be also nice to add if you have it
Response 7: This is also a very suitable request. Unfortunately, we only have a clinical picture for one case, which we added to the manuscript.
Comment 8: adding treatment done for each reviewed article is important in my views (surgical treatment/ chemotherapy/ radiotherapy etc)
Response 8: Thank you for this feedback. We agree that presenting the treatment for each case is important. Therefore, we also added this information in the table for the available cases. Unfortunately, this information is quite limited and as most of the cases presented were diagnosed before targeted therapy and immunotherapy were available, the treatment do not reflect the current standards and outcomes for cutaneous melanoma.
Comment 9: case presentation (of 2 cases), add size of lesion , workup for staging (what imaging), how follow-up was done, surgical margins (R classifications)
Response 9: We also appreciate this suggestion. Similar to your previous comment, we also added this pieces of information in the presentation of the cases.
Comment 10: enrich discussion with staging work-up imaging, and recommended type of follow up (local only/ PET/ or CT chest/ local MRI ?) , minimal recommended surgical margins, recommended treatment (surgical only / chemotherapy in metastatic cases or other )
Response 10: Thank you for this suggestion. We agree that the discussion section would benefit from this kind of information. Therefore, we added a new paragraph at the end of the manuscript adressing all these matters.
Reviewer 2 Report
Comments and Suggestions for Authors
The paper by Iliesiu et al., entitled “Primary Cutanoues Rhabdomyosarcomatous Melanomas: A Case Series and Literature Review,” presents a case series along with a literature review on rhabdomyosarcomatous melanoma. The authors detail two cases of rhabdomyosarcomatous melanomas that provide definitive evidence of myogenic differentiation. Overall, the paper is well-written and engaging, making it appealing to the journal's readers. However, I would like to raise the following minor concerns:
- It would be beneficial to include clinical pictures of the two cases.
- Could the authors clarify how they distinguish between DAB expression and melanin on immunohistochemistry?
3. The title contains a typographical error; "cutanoues" should be corrected to "cutaneous."
Author Response
We would like to thank the reviewer for accepting to review this manuscript and for the suggestions.
Comment 1: It would be beneficial to include clinical pictures of the two cases.
Response 1: This is indeed a very good suggestion. Unfortunately, we only have the clinical picture for one of the cases which we added to the manuscript.
Comment 2: Could the authors clarify how they distinguish between DAB expression and melanin on immunohistochemistry?
Answer 2: The distinction can be quite difficult sometimes and using red chromogen may be necessary. However, due to the very few pigmented cells, we could directly compare the staining pattern on the IHC slides with the histology slides.
Comment 3: The title contains a typographical error; "cutanoues" should be corrected to "cutaneous."
Answer 3: Thank you for this observation. We have modified the title.
Round 2
Reviewer 1 Report
Comments and Suggestions for Authors
Dear Authors,
Thank you for your revised manuscript and hard work and all explainations
congratulations !